# Advances in Percutaneous Patent Foramen Ovale Closure: From the Procedure to the Echocardiographic Guidance

**DOI:** 10.3390/jcm11144001

**Published:** 2022-07-11

**Authors:** Simona Sperlongano, Mario Giordano, Giovanni Ciccarelli, Giuseppe Bassi, Marco Malvezzi Caracciolo D’Aquino, Carmen Del Giudice, Gianpiero Gaio, Antonello D’Andrea, Adriana Postolache, Maurizio Cappelli Bigazzi, Giancarlo Scognamiglio, Berardo Sarubbi, Maria Giovanna Russo, Paolo Golino, Patrizio Lancellotti

**Affiliations:** 1Division of Cardiology, Department of Traslational Medical Sciences, University of Campania Luigi Vanvitelli, 80131 Naples, Italy; bassi.giuseppe93@gmail.com (G.B.); carmen.delgiu93@gmail.com (C.D.G.); paolo.golino@unicampania.it (P.G.); 2Division of Pediatric Cardiology, University of Campania Luigi Vanvitelli, Monaldi Hospital, 80131 Naples, Italy; mariojordi@msn.com (M.G.); gianpiero.gaio@hotmail.com (G.G.); 3Vanvitelli Cardiology Unit, Monaldi Hospital, 80131 Naples, Italy; ciccarelli.giovanni@gmail.com (G.C.); marcomalv2@hotmail.com (M.M.C.D.); mcappellibigazzi@gmail.com (M.C.B.); mgiovannarusso@gmail.com (M.G.R.); 4Department of Cardiology and Intensive Coronary Care, Umberto I Hospital, 84014 Nocera Inferiore, Italy; antonellodandrea@libero.it; 5Department of Cardiology, GIGA Cardiovascular Sciences, CHU Sart Tilman, 4000 Liege, Belgium; adriana.postolache@gmail.com (A.P.); plancellotti@chuliege.be (P.L.); 6Gruppo Villa Maria Care and Research, Maria Cecilia Hospital, 48033 Cotignola, Italy; 7Department of Cardiac Surgery, Anthea Hospital, 70124 Bari, Italy; 8Unit of Adult Congenital Heart Disease, Monaldi Hospital, 80131 Naples, Italy; giancascognamiglio@alice.it (G.S.); berardo.sarubbi@virgilio.it (B.S.)

**Keywords:** patent foramen ovale (PFO), transesophageal echocardiography (TEE), intracardiac echocardiography (ICE), Amplatzer PFO Occluder, Gore Cardioform Septal Occluder, NobleStitch

## Abstract

Percutaneous patent foramen ovale (PFO) closure by traditional, double disc occluder devices was shown to be safe for patients with PFO, and more effective than prolonged medical therapy in preventing recurrent thromboembolic events. The novel suture-mediated “deviceless” PFO closure system overcomes most of the risks and limitations associated with the traditional PFO occluders, appearing to be feasible in most interatrial septum anatomies, even if data about its long-term effectiveness and safety are still lacking. The aim of the present review was to provide to the reader the state of the art about the traditional and newer techniques of PFO closure, focusing both on the procedural aspects and on the pivotal role of transesophageal echocardiography (TEE) in patient’s selection, peri-procedural guidance, and post-interventional follow-up.

## 1. Introduction

Patent foramen ovale (PFO) is a cardiac congenital anomaly with high prevalence in the general population (20–30%) and often represents an incidental finding [1]. PFO can be implicated in the pathogenesis of several medical conditions, so called PFO-associated syndromes, including cryptogenic thromboembolism, decompression sickness, migraine, and arterial deoxygenation syndromes. In the absence of specific guidelines and high levels of evidence, European position papers have been drawn up, aiming at orienting the management of PFO and PFO-associated syndromes [1,2]. The current main indication for percutaneous PFO closure is a confirmed cryptogenic stroke, transient ischemic attack (TIA), or systemic embolism in patients aged from 18 to 65 years, with an estimated high probability of a causal role of PFO [2]. In this subset of patients, percutaneous PFO closure with implantable occluder devices based on an “umbrella-like” double disc design was shown to be more effective than prolonged medical treatment [3,4,5]. Moreover, percutaneous PFO closure is associated with a low morbidity, offering a safe and minimally invasive solution to definitely occlude PFO and reduce the risk of stroke.

Recently, a new percutaneous “deviceless” system based on surgical suture-mediated PFO closure has been introduced in interventional practice to overcome the potential issues related to the use of traditional devices [6].

The aim of the present review is to provide to the reader the state of the art about the techniques of PFO closure, from the traditional to the more recent ones, focusing both on the procedural aspects and on the role of transesophageal echocardiography (TEE) in patient’s selection, peri-procedural guidance, and post-interventional follow-up.

## 2. Percutaneous Closure of PFO by Traditional Devices

In 1975, King and Mills performed for the first time a percutaneous atrial septal defect closure in humans [7] by using an automatically opening “umbrella-like” device, further refined to the clamshell double-umbrella device, which had the advantage of being recapturable or removable up to the end of the procedure [7]. Due to the occurrence of late fractures of the metal arms [8], the device was withdrawn from the market. In the mid-1990s, the first Amplatzer Septal Occluder (Abbott, Chicago, IL, USA) was designed as a self-expandable, fully retrievable double disc device made of nitinol wire [9]. Nitinol is a metal alloy of nickel and titanium characterized by thermal memory. This property allows devices to be compressed at a small diameter into a delivery system. When the device is deployed, it comes back to its original configuration.

Then, the Amplatzer PFO Occluder was specifically designed for PFO closure [10], becoming the most widely investigated and used device for percutaneous PFO closure and atrial septal closure in general [11,12,13].

In recent decades, several dedicated devices and closure systems have been developed for the catheter-based therapy of PFO [4,5,11,12,14,15,16,17,18,19,20,21,22,23,24]. The most used are the double disc devices (the traditional ones).

### 2.1. Double Disc Devices

The Amplatzer PFO Occluder and the Gore Cardioform Septal Occluder (W.L. Gore and Associates; Flagstaff, AZ, USA) are the most used devices in clinical practice so far [2] (Figure 1).

The Amplatzer PFO Occluder, made from nitinol and polyester, is a self-expanding, non-self-centering, double disc device with a smaller left and a larger right atrial disc (except for the sizes 18 mm and 30 mm, in which the diameter is the same for both discs) and a central thin stem. The Amplatzer Multi-Fenestrated Septal Occluder—Cribriform (Abbott, Chicago, IL, USA) is also often implanted for PFO closure, and it differs from the Amplatzer PFO Occluder just for the presence of two equal size discs. Procedural success rates are close to 100%, and effective closure rates are as high as 95% at 6 months [12]. Thrombus formation on the device is exceedingly rare, as is clinically relevant new-onset atrial fibrillation.

The Gore Cardioform Septal Occluder is a non-self-centering, double disc device made out of a nitinol frame, covered by expanded polytetrafluoroethylene (ePTFE). Despite high success rates, a lower complete closure rate has been reported with this device [2]. On the other hand, it has been reported that the Cardioform device has the highest complete closure rate, based on a transcranial Doppler bubble study assessment for residual shunting [25]. However, no cases of cardiac erosion with this device are reported in the literature.

Amplatzer Septal Occluders (Abbott, Chicago, IL, USA) and Gore Cardioform ASD Occluder (W.L. Gore and Associates; Flagstaff, AZ, USA) are self-centering, double discs devices characterized by a central large waist which allows perfect device-centering. They are designed for atrial septal defect closure; however, they may be useful for closing PFO with specific anatomies.

A successful double disc devices implantation rate amongst all randomized clinical trials (RCTs) was achieved of 95.6%, with no or minimal residual shunt. These results support the fact that percutaneous PFO closure is a feasible and safe procedure, with no increase in serious adverse events. The most frequent late complications are device thrombosis, the incidence of which is 1.0–2.0%, and device embolism, which occurs at a rate of 0.9–1.3% [26]. A recent meta-analysis showed no significant differences between the device closure group and medical therapy group as regards the risk of serious adverse events and major bleedings [27,28]. Residual shunts, which are associated with increased rates of recurrent events, might be treated successfully with a second device [29,30,31]. New-onset atrial fibrillation (AF) and flutter are the most frequent arrhythmic complications, with an incidence of about 3.2% [32], possibly due to catheter manipulation, wire crossing into the left atrium, or stretching of the atrial wall with the device. However, AF commonly occurs within 45 days after device implantation, and in 76% of cases is a transient phenomenon without recurrence [32]. According to a recent retrospective cohort study on 1533 Ontarian patients, the incidence of AF after PFO closure was relatively low (6.26%) over a follow-up of about 8 years [33].

Currently, the following devices can also be used: Figulla Flex II PFO Occluder (Occlutech GmbH, Jena, Germany), Figulla Flex II UNI Occluder (Occlutech GmbH, Jena, Germany), Figulla Flex II ASD Occluder (Occlutech GmbH, Jena, Germany), CeraFlex PFO Occluder (Lifetech Scientific Corporation, Shenzhen, China), Ultrasept PFO Occluder (Cardia Inc., Eagan, MN, USA), Hyperion PFO Occluder—II (Comed B.V., Bolsward, The Netherlands), Nit-Occlud PFO (PFM Medical, Cologne, Germany), Amender PFO Occluder (Kewei Rising Medical Co., Ltd., Guangdong, China) and MemoPart PFO Occluder (Lepu Medical Technology Co., Beijing, China).

### 2.2. Procedural Steps

PFO closure is routinely performed as a day-case procedure, usually under TEE guidance.

A venous vascular access is obtained to perform the procedure. The femoral vein is the preferred vascular access; however, in rare cases, the femoral vein cannot be used due to bilateral thrombosis or specific anatomical variants (i.e., inferior vena cava interruption with azygos-continuation). In the latter cases, the internal jugular vein, the axillary/subclavian vein or the suprahepatic veins may be used for the procedure [34,35,36].

PFO is usually crossed with a 5–6 Fr multipurpose catheter and a standard 0.035-inch guidewire from the right atrium. From the inferior vena cava, the Eustachian valve directs the guidewire into the PFO tunnel. However, in some cases, a stiff tunnel prevents the smooth crossing of the guidewire. In such cases, it might be used as a hydrophilic guidewire to cross the tunnel or a smaller catheter (4 Fr multipurpose or JR) may be wedged into the tunnel with subsequent crossing after the initial thenting of the septum primum. Subsequently, a stiff guidewire can be placed in a left pulmonary vein (preferably the upper one to have a better alignment with interatrial septum). Often, an evaluation of the septum primum raising and the size of the consequent left-to-right shunt is enough to choose the device to use. However, balloon sizing of the tunnel may allow PFO stretching with a better device selection [37]. Usually, a tunnel stretch with a diameter ≤12 mm is addressed to a non-self-centering device with a maximum disc diameter of 25 mm, while a larger stretch diameter is addressed to bigger devices.

A long sheath is pushed beyond the PFO tunnel, the left-disc of the device is opened in the left atrium, pulled-back to the interatrial septum and then the right disc is opened in the right atrium and pushed to the PFO entry. Before releasing the device, it is crucial to ensure that:-The device has well hugged the septum primum, particularly the septum secundum;-There is no interference with the atrioventricular valves or the other intracardiac structures-There are no patent accessory fenestrations

When the device is well-deployed, the cranial left anterior oblique fluorscopic view shows a typical pattern of the prosthesis with the upper edges of the discs well separated and the lower ones very close, the so-called “Pacman sign”. In fact, the thick septum secundum divided the two discs at the superior vena cava outlet, whereas the thin septum primum allows greater disc compacting at the inferior vena cava outlet [38]. The device’s stability may be checked with a push-and-pull maneuver it is released (Figure 2).

### 2.3. Role of TEE

#### 2.3.1. TEE before PFO Closure: Characterization of the PFO and Decision Making

TEE provides a detailed visualization of the interatrial septum and other relevant structures, and it can show the shunt across the septum by using contrast (c-TEE). However, the accuracy and sensitivity of c-TEE in diagnosing PFO are inferior to other techniques, probably due to the inability to perform an adequate Valsalva maneuver during examination [39,40,41]. The imaging method with the highest sensitivity in detecting the right-to-left interatrial shunt is contrast-enhanced transcranial Doppler (c-TCD), which is the first-line investigation in PFO diagnostic workup. TEE, rather than a diagnostic role, plays a key role in defining the anatomy of the interatrial septum, aiming at stratifying the risk and assessing the suitability for device closure.

TEE imaging of the interatrial septum begins with the transverse plane at 0° [42]. The interatrial septum should be swept from the superior vena cava down through the fossa ovalis up to the inferior vena cava and the coronary sinus at a high-, mid-, and low-esophageal level, respectively. Fossa ovalis, which is the region where the “septum primum” is exposed on the right atrium, is easy to recognize at mid-esophageal level, since it is thinner than the “septum secundum”. From 0°, a slow continuous sweep of the interatrial septum should be made with increments of 10–20° [43]. At 90° the interatrial septum is assessed along its longitudinal (supero-inferior) plane, always advancing the probe from the superior vena cava to the mid-esophageal level towards the inferior vena cava. The 45° view at the mid-esophageal level displays the relationship of the interatrial septum and fossa ovalis with the aortic valve and aortic root (Figure 3). The opening of the PFO in the left atrium can be seen in each of the above-mentioned views from the high- or mid-esophageal levels. For each plane, it may be useful to visualize the entire interatrial septum with both 2-dimentional (2D) gray-scale imaging and 2D color Doppler imaging to avoid missing other atrial defects (e.g., small fenestrations in the fossa ovalis or other larger defects). Reducing the color Doppler scale (20–30 cm/s) can be helpful to optimize blood flow visualization across the PFO. Then, intravenous agitated saline contrast should be administered to confirm that the passage occurs across the PFO (bubble test). In the presence of a PFO, a right-to-left shunt should be observed early after a correctly performed Valsalva maneuver (within 3–5 beats from the release). The effectiveness of the Valsalva maneuver is evidenced echocardiographically by the leftward bulging of the “septum primum”, denoting that the right atrial pressure is higher than the left atrial. A real-time three-dimensional (3D) analysis may complement the 2D TEE, allowing a clearer understanding of PFO morphology and its relationship with surrounding structures. In particular, real time 3D TEE offers improved spatial resolution for the features of complex PFOs (e.g., aneurismatic interatrial septum, multiple fenestrations). The best 3D imaging of the interatrial septum should be performed in “3D zoom” mode, acquiring the bicaval view (90–110°), where the atrial septum is perpendicular to the ultrasound beam.

Table 1 contains the anatomical features of the interatrial septum that should be assessed by TEE for a patient’s risk stratification and interventional treatment. Atrial septal aneurysm, moderate-to-severe shunt, atrial septal hypermobility, and large PFO size were found to be strongly associated with a causal role of PFO in left circulation thromboembolism. [4,14,44,45,46,47]. Hence, their presence is currently an indication for percutaneous PFO closure. Prominent Eustachian valve, Chiari network, and long PFO tunnel are other despite the lower level of scientific evidence [48,49].

#### 2.3.2. TEE during PFO Closure: Intra-Procedural Guidance

During the percutaneous procedure of PFO closure, TEE is used in combination with fluoroscopy, since it displays long segments of catheters and wires and their relationship with surrounding structures. TEE guidance is needed to cross the PFO, position the closure device, verify if its location is proper, and ensure its stability and efficacy [43]. Closure device stability is evaluated by the so-called “wiggle maneuver”. During the “wiggle maneuver” the occluder, which is deployed but still connected to the delivery cable, is pushed and pulled before release to make sure that neither the right nor the left disk slips into the contralateral chamber [50]. TEE imaging also allows the post-closure monitoring of adjacent structures, in particular mitral valve competency and coronary sinus patency, and the detection of complications immediately following device deployment, including left atrial appendage perforation, intracardiac thrombus formation, or early device embolization.

During the procedure, the “3D zoom” mode can be used to visualize the position of guide wires, catheters, and devices in real time. The PFO closure device position and its spatial relationship with adjacent native structures can be displayed at any time and from each side by 3D TEE.

#### 2.3.3. TEE after PFO Closure: Post-Procedural Follow-Up

A TEE 6 months after PFO device implantation is recommended to confirm the correct device localization and exclude late complications. C-TEE allows us to visualize residual right-to-left shunts around or within the device. Shunts following the PFO closure procedure generally disappear or decrease overtime after device endothelialization. However, in some patients they persist and can be seen many months after the procedure. Serial TEEs are useful for monitoring the degree of passage and for identifying large and persistent shunts, which may benefit from re-intervention. TEE is more sensitive than transthoracic echocardiography in detecting thrombus formation on device surface, a complication mostly observed within the first month after device implantation [51,52]. TEE is also the tool of choice to diagnose the rare but serious infective endocarditis, following PFO closure device implantation, revealing the presence of vegetations on the left or the atrial side of the device [53]. The erosion of adjacent cardiac structures, including right or left atrial roof and aortic root, and the device embolization, are other possible late complications following percutaneous device closure of PFO, which can be detected by TEE.

#### 2.3.4. Intracardiac Echocardiography

Intracardiac echocardiography (ICE) is an alternative to TEE during PFO closure, which can directly diagnose PFO and continuously guide the procedure [54]. It is typically performed by an interventional cardiologist using a 7–10 F catheter that is introduced via a second femoral venous sheath [55]. ICE has high image resolution, allowing an accurate assessment of PFO size, position, and edges at different angles [56]. Two orthogonal views (the transverse section at the aortic valve level and the longitudinal section of the four-chamber plane) are generally acquired to measure the defect and guide the deployment of the device. ICE can also be used to detect residual shunting immediately after percutaneous PFO closure, and to monitor acute complications, such as thrombus formation and pericardial effusion [56]. ICE has the advantage of avoiding the need for general anesthesia and intubation, and their associated risks. Moreover, it reduces the duration of radiation exposure (X-ray is only used before the catheter arrives at the right atrium), which is crucial for children and pregnant women. However, ICE use is limited by the cost of a single-use probe, the need for specific training, and the risk of complications related to the femoral puncture.

## 3. Percutaneous Closure of PFO with Complex Anatomy

In most cases, PFO has a usual anatomy in which a non-self-centering device can achieve effective complete closure. However, in a minority of cases, the interatrial septum appears with some specific features that may complicate the procedure. The most common complex anatomies of PFO are: aneurismatic interatrial septum, multiple accessory fenestrations, long and stiff PFO tunnel, lipomatous and hypertrophic septum secundum, exuberant Eustachian valve (Chiari network) and malalignment of the interatrial septum. These features may be isolated or associated with each other.

### 3.1. Aneurismatic Interatrial Septum

The interatrial septum is defined as “aneurismatic” when it has a base width ≥15 mm and an excursion into either the left or the right atrium >10 mm. [57]. Olivares-Reyes et al. introduced a descriptive classification of the interatrial septal aneurysms, examining the direction of septal excursion: type 1R (dextro-convex), the aneurysm protrudes into the right atrium; type 2L (left-convex), the aneurysm protrudes into the left atrium; type 3RL, the aneurysm has a bidirectional waving with a prevalence of the right atrium excursion; type 4LR, the aneurysm has a bidirectional waving with a prevalence of the left atrium excursion; type 5, the aneurysm has a completely bidirectional movement without a prevalent excursion (Figure 4) [58]. The Olivares-Reyes classification is useful to describe the aneurysm even if the excursion direction does not influence the percutaneous procedure. Furthermore, the excursion of an aneurism is not a “fixed-feature” of interatrial septum, since it depends on the hemodynamic status of the atria.

In some cases, a “hypermobility” of the interatrial septum in the absence of the aforementioned criteria may be detected. In this case, the name “aneurism” cannot be used and the term “waving (or hypermobility) of the atrial septum” is preferred.

In patients with cerebrovascular accident, aneurismatic interatrial septum was found to be associated with PFO in 49% of cases [59].

In cases of percutaneous closure of the aneurismatic interatrial septum, balloon sizing of PFO allows us to determine the “stretching capability” of the tunnel. When the “stretch diameter” is ≥13 mm, the self-centering devices are preferred to non-self-centering devices, since they have a lower risk of residual shunt at follow-up [60,61]. The wide waist of self-centering devices allows complete “filling” of the large PFO tunnel, avoiding the chance of inter-discal residual shunt.

### 3.2. Multiple Accessory Fenestrations

The presence of one or more multiple accessory fenestrations of the interatrial septum might significantly complicate the procedure. In fact, a percutaneous closure of PFO is considered “effective” only if every site of potential right-to-left shunt is completely closed.

Multiple accessory fenestrations and aneurysm of the interatrial septum often coexist; however, there may be cases in which multiple fenestrations are present within a non-aneurysmatic septum (Figure 5). The prevalence of small additional atrial septal defects in patients with PFO and cerebrovascular events is about 11% [59].

The presence of multiple sites of interatrial shunt might sometimes require the implantation of multiple devices (Figure 6) [62]. In the past, the presence of >2 fenestrations was considered a relative contraindication to percutaneous closure [63]. However, nowadays, the layout of the interatrial septum and of the multiple fenestrations is the main feature to study to plan the procedure [64]. The procedural goal is to cross and close the largest defect. In some cases, the discs of the device can cover the accessory fenestrations and/or the tunnel of PFO. However, in other circumstances, the defects are too far apart and the use of two or more prostheses may be necessary. When there is not a predominant defect but multiple small fenestrations of interatrial septum, the preferred approach is to cross the central one and to implant a non-self-centering device to cover the peripheral fenestrations with the discs of the device. In this case, the choice of device size is dependent on the distance of the peripheral fenestrations from the central one [65].

### 3.3. Long and Stiff PFO Tunnel

A PFO tunnel is considered long when the overlap between the septum primum and secundum is ≥8 mm. Its prevalence is about 55% among patients with cerebrovascular accidents [59].

A long tunnel usually does not complicate percutaneous closure, since the discs of the device are able to press the thin septum primum, achieving adequate compaction of the device and the interatrial septum. However, in some circumstances, the septum primum may be very stiff. The suspect of a stiff tunnel arises when it opens <4 mm after the deployment of a stiff wire. In this case, the device fails to compact and appears “elongated”. The device should not be released in this condition because of the high risk of both embolization and residual shunt, particularly. Various techniques are proposed to achieve an adequate result in this subset.

The creation of a “controlled iatrogenic” fenestration at the entrance to the tunnel is the most used technique. The trans-septal puncture can create a little fenestration close to the opening of the PFO tunnel. A non-self-centering device is deployed within this fenestration so that the discs cover (“crush”) the PFO tunnel, obtaining its complete closure (Figure 7) [66]. This technique is useful and widespread, even if adds the risk of the trans-septal puncture to the procedure.

The balloon pull-back technique (from the left to the right atrium, such as a Rashkind atrioseptectomy) makes the PFO tunnel incompetent, resulting in an atrial septal defect with basal left-to-right shunt. A balloon wedge catheter is advanced to the left atrium, inflated with diluted contrast, and pulled back through the PFO to the right atrium. This technique can increase the compliance of the tunnel with consequent correct deployment of the device [67]. The limit of the technique is related particularly to the rigidity of the septum primum in adult patients, which might be ineffective in this approach.

Controlled balloon “angioplasty” of the tunnel is a safe and feasible technique to use in these cases. A PFO angioplasty with non-compliant high-pressure balloons (diameter 10–12 mm) can shorten and widen the tunnel. After angioplasty, the tunnel appears less rigid, and the device may be implanted effectively [68].

### 3.4. Hypertrophy of the Septum Secundum

A septum secundum with a thickness of ≥7 mm is defined “hypertrophic” (Figure 8). A mild or moderate hypertrophy usually does not complicate the percutaneous closure. However, when the hypertrophy is severe (≥15 mm), the risk of procedural failure is high. In this circumstance, common devices are not able to catch the hypertrophic septum secundum.

The use of ventricular septal defect occluders (Amplatzer Muscolar VSD Occluder, Abbot, Plymouth, MN, USA) may help to achieve the closure of PFO. In fact, the Amplatzer Muscolar VSD Occluder has a longer (taller) transverse waist, increasing the chances of catching the hypertrophied septum secundum [69].

### 3.5. Redundant Eustachian Valve and Chiari Network

The prevalence of the prominent Eustachian valve of about 18% was found among patients with PFO and cerebrovascular events [59].

A redundant Eustachian valve (or a Chiari network) may interfere with the compaction of the right disc to the interatrial septum. In this case, it may be useful to use a pigtail catheter to engage and pull-back the Eustachian valve toward the inferior vena cava. In this way, the right disc is able to advance to the atrial septum without interferences [70]. This technique is useful in the case of an exuberant Eustachian valve; however, it is ineffective when there is a Chiari network with a wide attachment to the septum. In this circumstance, the device is unable to trap the anterior (retro-aortic) rim of the tunnel, making the percutaneous closure unfeasible [71].

### 3.6. Malalignment of Interatrial Septum

The malalignment of the interatrial septum (or “spiral septum” or “double atrial septum”) is characterized by significant malalignment between the septum secundum and the flap valve with the presence of an associated “accessory attachment” on the left-atrium side of interatrial septum. This condition is usually associated with an atrial septal defect even if it may be detected in the context of a PFO, with a prevalence of about 10% among patients with cerebrovascular accident [59,72].

The presence of a malalignment of the intertrial septum may interfere with device deployment. The left-sided “accessory attachment” prevents the left disc from adequately reaching the atrial septum. A counterclockwise rotation of the device could overtake the “accessory attachment”, achieving the adequate deployment of the device. However, this condition increases the risk of device embolization and aortic root erosion [72].

## 4. Percutaneous Closure of PFO by Novel Devices: The NobleStitch

Recently, a new “deviceless” system has been developed for PFO closure, which uses a stitch technology to suture the septum primum and secundum, producing an “S-shape” closure of the PFO [6]. The first experience with this device was recently reported in the NobleStitch (Kardia, Milan, Italy) EL Italian registry and included 192 patients [6,73]. Procedural success was achieved in 96% of patients, but at a mean follow-up of 200 days, an exceedingly high rate of residual shunt was found (significant right-to-left shunting in 11%) [6]. Several advantages can be hypothesized in favor of the NobleStich system compared to traditional devices. The NobleStitch system is a fluoroscopic-guided procedure, and it may be performed without anesthesiological and echocardiographic support. It does not require the implantation of a permanent device (“deviceless” intervention), minimizing the risk of atrial chamber obstruction or device-related infection, abolishing the need of prolonged double antiplatelet therapy after the procedure, and removing the potential allergenic effect due to the nickel mesh. Moreover, the NobleStitch system guarantees no preclusion to future percutaneous left side heart interventions, which require interatrial septum puncture (e.g., mitral valve repair, atrial fibrillation ablation, left atrial appendage occlusion).

Although suture mediated PFO closure represents a good alternative for PFO closure when compared to classic device closure, there is still a lack of information about the long-term follow-up of this procedure and its real effectiveness.

### 4.1. Procedural Steps

The NobleStitch EL system consists of two dedicated suture delivery catheters (NobleStitch S and NobleStitch P) to capture and suture the septum secundum and the septum primum, using a polypropylene suture which produces an “S-shape” closure of the PFO. A third element, the KwiKnot™ catheter (HeartStitch, Inc., Fountain Valley, CA, USA), is advanced over the septum secundum and septum primum sutures to approximate both septa, achieving closure by securing the stitch.

Through the right femoral vein, a multipurpose is advanced to cross PFO with a 0.032” wire placed in the left upper pulmonary vein. Then, the multipurpose catheter is exchanged for a straightened 14 Fr Mullins sheath, and a 0.018” wire is advanced into the distal superior vena cava. Sizing balloon of PFO is performed during contrast injection to determine the anatomy of the septum secundum and the septum primum. The PFO anatomy is considered favorable for NobleStich closure when an overlap between the septum primum and secundum is detected during balloon inflation. The NobleStitch S and P catheters are sequentially advanced to suture the septum secundum and the septum primum, respectively. During this maneuver, special care must be taken to puncture the septum primum at the nadir of the septum secundum. The system is, then, removed along with the wire, leaving the sutures free. The suture ends are gently pulled to bend the septum primum towards the right atrium and close the PFO. Keeping the tension on the sutures, the KwiKnot catheter is, finally, used to advance and release a radiopaque polypropylene knot on the right side of the interatrial septum and to cut the proximal suture (Figure 9).

### 4.2. Role of TEE

TEE plays a key role in the selection of patients undergoing percutaneous suture-mediated closure of PFO. Classical tunnel-like PFO is the ideal anatomy for closure by Noblestitch. Conversely, atrial septal defects PFO-like (very short tunnels with left-to-right shunt) and PFOs with multifenestrated interatrial septum are non-suitable anatomies, since they predict poor procedural success [74,75]. Finally, atrial septal aneurysms are to be evaluated case by case. Therefore, during pre-procedural evaluation by TEE imaging, particular attention should be paid to the presence of left-to-right shunts along the entire interatrial septum, and to the septum primum mobility. Morphological analysis of the interatrial septum, performed at mid-esophageal level (at about 0°, 45°, and 90° at least) by both 2D grayscale and 2D color Doppler imaging, needs to be completed by the bubble test at rest and after Valsalva maneuver. A Valsalva maneuver without contrast injection can be helpful to evaluate the aneurysm and the septal excursion.

After the procedure, a new contrast injection at rest and after Valsalva is recommended to verify the effectiveness of the percutaneous closure.

Table 2 summarizes the main characteristics of the NobleStitch^TM^ EL.

## 5. Conclusions

Percutaneous PFO closure by traditional occluder devices (“umbrella-like” double disc prosthesis) was shown to be an effective and safe therapy in patients with PFO and increased risk of recurrent thromboembolic events. TEE plays a pivotal role in the pre-procedural evaluation of the suitability for intervention, intra-procedural guidance in combination with fluoroscopy, and post-procedural assessment of effectiveness and possible complications.

The novel suture-mediated “deviceless” PFO closure system proved to be feasible in most interatrial septum anatomies, overcoming most of the limitations and risks associated with the traditional PFO occluders. It is reasonable to assume that this new technique will be increasingly used and, in the not-too-distant future, it will help to broaden the indications for PFO closure, if its long-term effectiveness and safety can be confirmed.

## Figures and Tables

**Figure 1 jcm-11-04001-f001:**
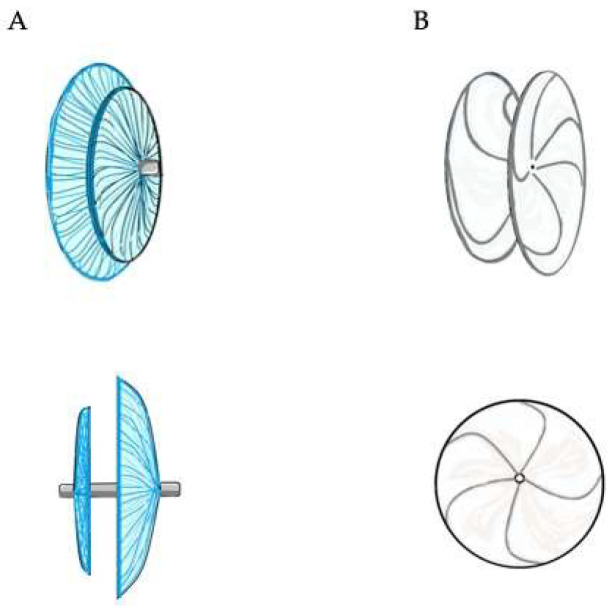
Double disc devices. The Amplatzer PFO Occluder (panel (**A**) on the left) is a self-expandable device consisting of two atrial discs, of which the right is larger than the left one, linked together by a short connecting waist. It is made from a nitinol wire mesh and a thin polyester fabric, securely sewn to each disc by a polyester thread, in order to increase its closing ability. The Gore Cardioform Septal Occluder (panel (**B**) on the right) consists of two atrial discs made from a nitinol wire frame with a platinum filled core. The two discs are covered with a thin, soft and conformable, expanded polytetrafluoroethylene (ePTFE) membrane.

**Figure 2 jcm-11-04001-f002:**
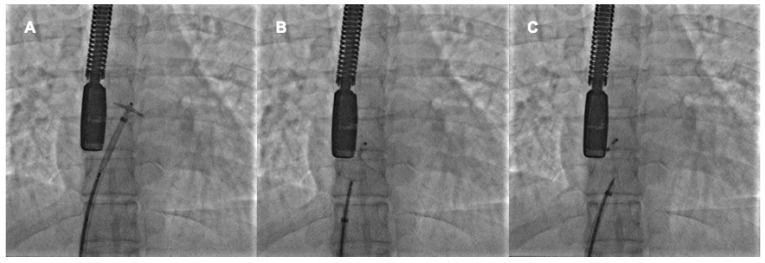
Percutaneous PFO closure by Amplatzer PFO Occluder 25 mm. The left disc is opened in the left atrium and pulled back to the interatrial septum (**A**). The right disc is opened in the right atrium and pushed towards the interatrial septum (**B**). After the push-and-pull manoeuvre, the device is released (**C**).

**Figure 3 jcm-11-04001-f003:**
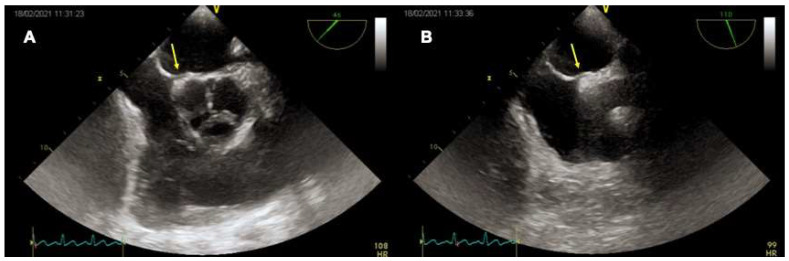
TEE imaging of the interatrial septum. Mid-esophageal 45° (**A**) and 90° (**B**) views of a tunnel-like PFO (yellow arrows).

**Figure 4 jcm-11-04001-f004:**
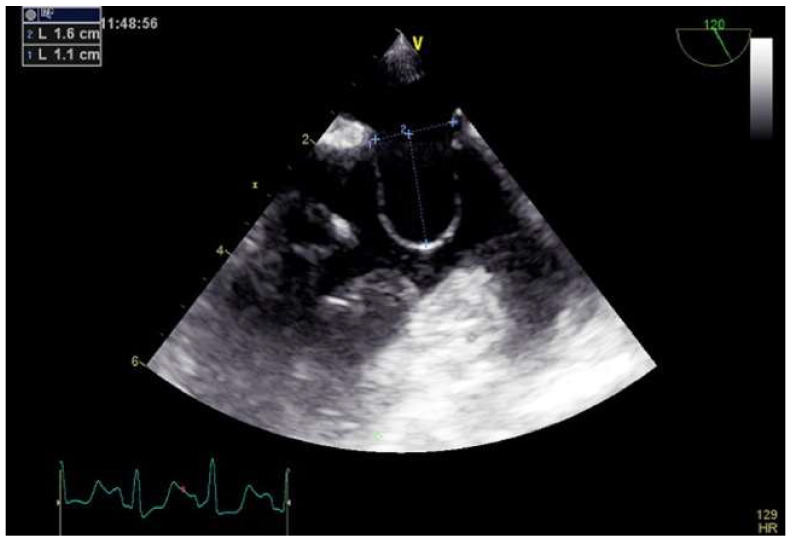
Type 1R (right-convex) interatrial septum aneurism.

**Figure 5 jcm-11-04001-f005:**
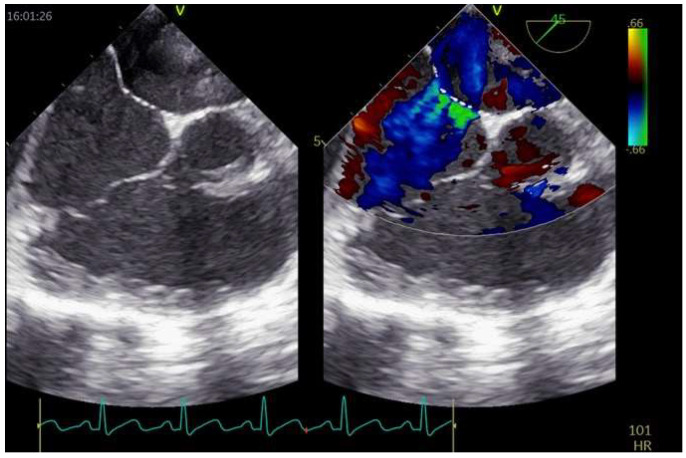
Non-aneurismatic interatrial septum with multiple fenestrations.

**Figure 6 jcm-11-04001-f006:**
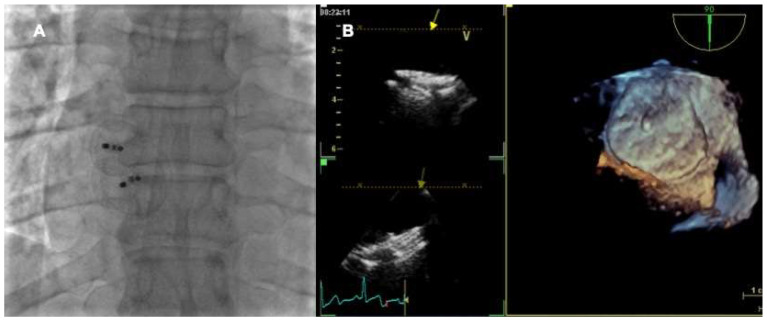
Multiple devices percutaneous implantation. Fluoroscopic (**A**) and 3D transesophageal (**B**) views of a PFO with a single accessory fenestration undergone percutaneous closure with two Amplatzer PFO Occluder devices in partial overlap.

**Figure 7 jcm-11-04001-f007:**
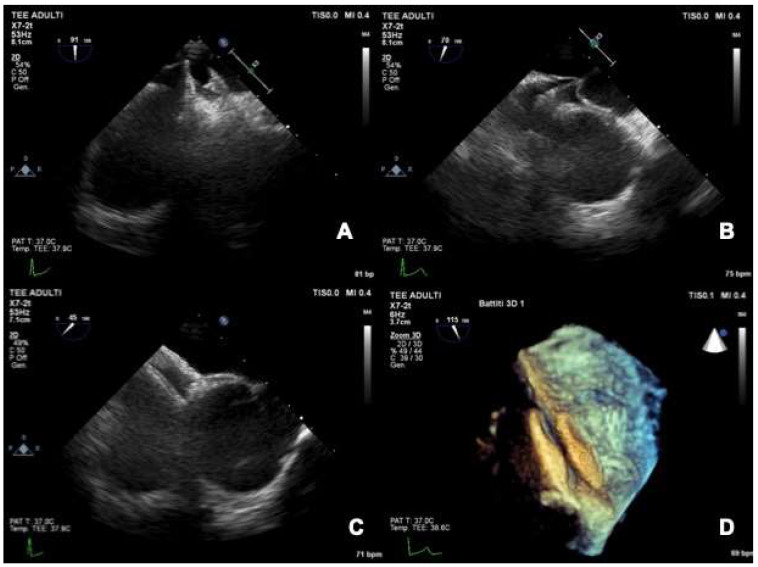
Percutaneous closure of a long and stiff PFO tunnel. A long and stiff tunnel does not allow a correct compaction of the two discs (**A**). The device is pulled back and a trans-septal puncture is performed at the entrance of the tunnel (**B**). The device is deployed again through the trans-septal puncture, with adequate discs compaction and complete closure of the patent foramen ovale (**C**,**D**).

**Figure 8 jcm-11-04001-f008:**
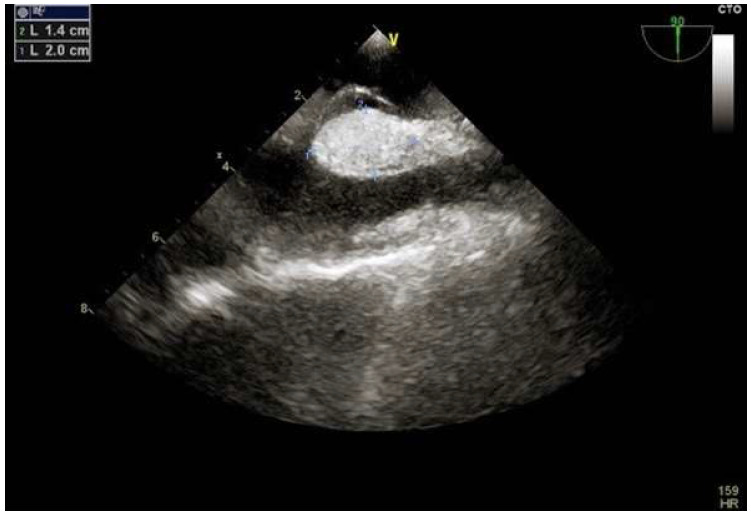
Hypertrophy of the septum secundum.

**Figure 9 jcm-11-04001-f009:**
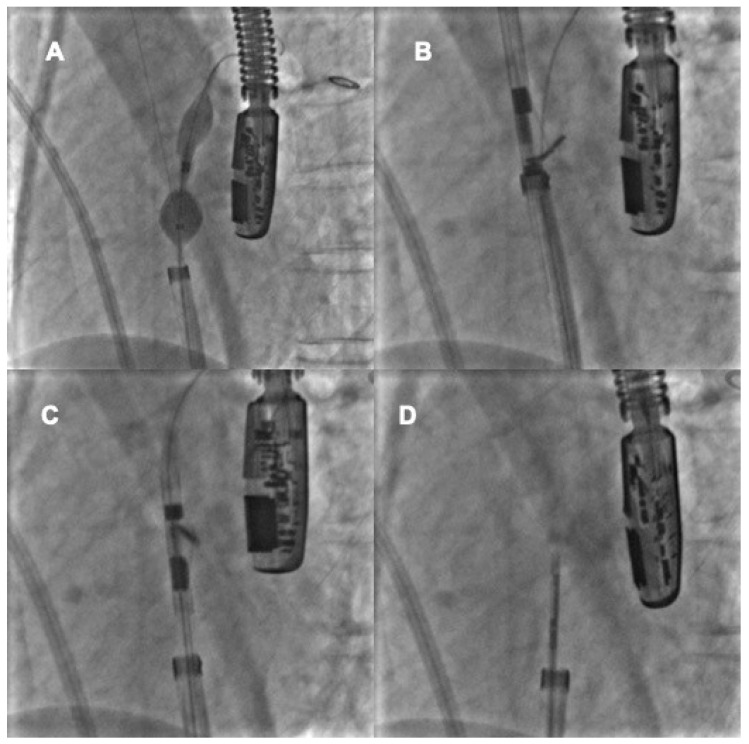
Percutaneous PFO closure by NobleStitch EL system. Sizing balloon of the PFO is performed to determine the anatomy of the septum secundum and the septum primum during contrast injection (**A**). NobleStitch S and P catheters are sequentially advanced to suture the septum secundum and the septum primum, respectively (**B**,**C**). The delivery system is removed, and the suture ends are gently pulled to close the PFO. Finally, maintaining the tension on the sutures, the KwiKnot catheter is used to advance and release a polypropylene knot on the right side of the interatrial septum, and to cut the proximal suture (**D**).

**Table 1 jcm-11-04001-t001:** Interatrial septum variables to be assessed by transesophageal echocardiography for decision making.

- PFO location - Length of the PFO tunnel - Width of the PFO tunnel opening - Patency of foramen ovale at rest and after Valsalva manoeuvre by color Doppler and by - injecting contrast - Distance between PFO and aortic root, superior and inferior vena cava, right - upper pulmonary vein, atrio-ventricular valves, and posterior atrial wall (rims) - Atrial septal mobility - Concomitant atrial septal aneurysm - Septal length - Thickness of the septum secundum
- Additional atrial septal defects - Pulmonary venous anatomy - Left atrial appendage orientation - Eustachian valve and/or Chiari network

PFO: patent foramen ovale.

**Table 2 jcm-11-04001-t002:** Characteristics of NobleStitch^TM^ EL.

**Indication of use**	PFO closure in patients with systemic thromboembolism and high probability of PFO causal role
**Type of technology**	Suture-mediated “deviceless” technology
**Type of access**	Percutaneous (right femoral vein preferentially)
**Role of TEE**	- Pre-procedural selection of patients (evaluation of interatrial septum morphology) - Post-procedural assessment of percutaneous closure effectiveness
**Advantages**	- Risk reduction in potential early and late complications due to the presence of the double-disc device (e.g., device dislodgment, migration or embolization, atrial wall erosion, infection, thrombosis, induction of arrhythmias, etc.) - No need for prolonged dual antiplatelet therapy after the procedure - Possibility of use in patients allergic to nickel - No obstacle to future transseptal puncture and left-sided interventions (e.g., left atrial appendage closure, arrhythmia ablation, mitral valve interventions, etc.) - TEE and general anesthesia not mandatory
**Disadvantages**	Non-feasible in some interatrial septum anatomies (e.g., PFOs atrial septal defects-like, PFOs with multifenestrated interatrial septum)
**Success rate ***	96%

PFO: patent foramen ovale; TEE: transesophageal echocardiography, * According to the NobleStitch EL Italian Registry by Gasperdone et al. [6].

## Data Availability

Not applicable.

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
