# Peer review of "Advances in Percutaneous Patent Foramen Ovale Closure: From the Procedure to the Echocardiographic Guidance"

_jcm, 2022, doi:10.3390/jcm11144001_

Round 1
Reviewer 1 Report
The authors present a thorough, but excellent overview of PFO closure and the role of echocardiography. The manuscript would be a great contribution to the field. I only have a few minor comments:
- With regards to section 3: percutaneous closure of PFO with complex anatomy. It would be useful to cite the prevalence/incidence of these complex cases, if these numbers are available.
- Recent work has shown a long-term association between PFO closure devices and the incidence of atrial fibrillation. The sentence in line 113-115 should be updated with information from both PMID: 35429648 and the associated editorial PMID:35643300 which gives useful insights in future research objectives. Even though this review provides mostly procedural insights, the AF association is clinically relevant given the predominant indication for PFO closure is TIA/stroke.
- Minor grammatical language edits are required.
Author Response
Please, see the attachment.

Reviewer 2 Report
This review outlines the techniques of PFO closure, from the traditional to the more recent ones, focusing both on the procedural aspects and on the role of transesophageal echocardiography.
Major Concerns:
1. What was done to ensure the review is comprehensive and that no critically important studies were missed? What were the inclusion criteria for a study to be included in the submitted review? Did the authors adopt a systematic review methodology?
2. The methods of the literature review are missing. How did the Authors structure the literature research and why? How did you select literature for inclusion?
3. There is no analysis or discussion in this review to help the reader to decide what factors should go into determining which technique should be used.
4. Compare the mechanisms and help the reader understand their similarities and differences and the factors to consider when comparing methods. For example, is one considered the best right now? Several, but it depends on goals/context?
Author Response
Please, see the attachment.
